# Non-Coding RNAs in Cartilage Development: An Updated Review

**DOI:** 10.3390/ijms20184475

**Published:** 2019-09-11

**Authors:** Ehsan Razmara, Amirreza Bitaraf, Hassan Yousefi, Tina H. Nguyen, Masoud Garshasbi, William Chi-shing Cho, Sadegh Babashah

**Affiliations:** 1Department of Medical Genetics, Faculty of Medical Sciences, Tarbiat Modares University, Tehran P.O. Box 14115-111, Iran; 2Department of Molecular Genetics, Faculty of Biological Sciences, Tarbiat Modares University, Tehran P.O. Box 14115-111, Iran; 3Department of Biochemistry and Molecular Biology, LSUHSC School of Medicine, New Orleans, LA 70112, USA; 4Department of Clinical Oncology, Queen Elizabeth Hospital, Kowloon, Hong Kong

**Keywords:** chondrogenesis, non-coding RNAs, lncRNAs, microRNAs, cartilage development

## Abstract

In the development of the skeleton, the long bones are arising from the process of endochondral ossification (EO) in which cartilage is replaced by bone. This complex process is regulated by various factors including genetic, epigenetic, and environmental elements. It is recognized that DNA methylation, higher-order chromatin structure, and post-translational modifications of histones regulate the EO. With emerging understanding, non-coding RNAs (ncRNAs) have been identified as another mode of EO regulation, which is consist of microRNAs (miRNAs or miRs) and long non-coding RNAs (lncRNAs). There is expanding experimental evidence to unlock the role of ncRNAs in the differentiation of cartilage cells, as well as the pathogenesis of several skeletal disorders including osteoarthritis. Cutting-edge technologies such as epigenome-wide association studies have been employed to reveal disease-specific patterns regarding ncRNAs. This opens a new avenue of our understanding of skeletal cell biology, and may also identify potential epigenetic-based biomarkers. In this review, we provide an updated overview of recent advances in the role of ncRNAs especially focus on miRNA and lncRNA in the development of bone from cartilage, as well as their roles in skeletal pathophysiology.

## 1. Introduction

Chondrocytes play a central role in endochondral bone development, which is a necessary step for skeletal development. During this complex process, chondrocytes undergo various stages of sequential differentiation, including proliferation and hypertrophy, which are regulated by various transcription factors [1] (Figure 1A). Chondrogenesis begins with the formation of anlagen, a cartilage intermediate that marks the first step of endochondral ossification (EO) during skeletal development [2]. Mesenchymal cells play a key role in embarking on this process; these cells are derived from three sources: neural crest cells of the neural ectoderm [3], the sclerotome of the paraxial mesoderm [4], and the somatopleure of the lateral plate mesoderm [2]. These cells respectively develop into the craniofacial bones, axial skeleton, and long bones. Though genetic, epigenetic, and environmental factors are known to regulate the formation of the skeleton from chondrogenesis, novel technologies allow for a deeper understanding of the importance of epigenetic influences.

The vast majority of transcribed mRNA is not translated into protein, and these sequences are known as non-protein-coding RNAs (ncRNAs). ncNAs are frequently included among the mechanisms of epigenetic control and are categorized into either small RNAs (<200 bp) or long RNAs (>200 bp) [5]. One of the more well-known small ncRNAs is microRNA (miRNA), which attenuates protein synthesis by binding to the 3́ untranslated region (3́ UTR) of target mRNAs. Long non-coding RNAs (lncRNAs) are long non-translational RNAs that are involved in many pivotal biological processes and can function as decoys, scaffolds, and enhancer RNAs [6].

In this article, we discuss the various functions of miRNA and lncRNA in different stages of chondrogenesis, as well as their roles and regulatory potential in chondrocyte development.

## 2. Cartilage Development

Cartilage is an elastic tissue composed of specialized cells, primarily chondrocytes, which are responsible for secreting a large amount of collagenous extracellular matrix (ECM). Cartilage can be categorized into three different types: elastic cartilage, hyaline cartilage, and fibrocartilage, which differ in content or amount of collagen and proteoglycan [6]. EO starts with mesenchymal cell condensation which is followed by differentiation into chondrocytes. These chondrocytes produce cartilage by secreting an abundant ECM mixture of collagen type II alpha 1 (COL2A1) and aggrecan (ACAN) [7]. The differentiated cell then goes through steps of differentiation into proliferating, hypertrophic, and mature chondrocytes (Figure 1). Ultimately, these chondrocytes undergo apoptosis and are replaced by bone [8].

Cell-specific markers play a crucial role in chondrocyte differentiation. *COL2A1* and *ACAN* are expressed in proliferating chondrocytes. Indian hedgehog (Ihh) and parathyroid hormone 1 receptor (PTH1R) are expressed in pre-hypertrophic chondrocytes. Collagen type X, alpha 1 (*COL10A1*) and matrix metalloproteinase 13 (*MMP13*) are expressed in hypertrophic chondrocytes (Figure 1A) [9]. Also, various transcription factors play role in the differentiation stages; for example, at the beginning of chondrocyte differentiation, the SRY-box containing gene (Sox) family, including SOX5, SOX6, and SOX9 regulate *COL2A1* gene expression [10,11,12] (Figure 1B). Any defects in the Sox family lead to severe defects in skeletal development including skeletal dysplasia [13].

Several proteins are important in chondrocyte hypertrophy, the second phase of chondrocyte differentiation (Figure 2). These proteins include myocyte-specific enhancer factor 2C (MEF2C), Forkhead box (FoxA), and the runt-related (Runx) transcription factor family proteins [14,15,16]. Runx2 plays an essential role in regulating the expression of Ihh by binding to the promoter region, as evident in knock out mice for Runx2 and Runx3 chondrocyte hypertrophy is delayed [14].

Runx2 and SOX9 are two pivotal transcriptional regulators essential for hypertrophic maturation and articular cartilage formation, respectively [17,18]. For example, the activation of Runx2 by transforming growth factor β1 (TGFβ1) and bone morphogenetic protein (BMP) can differentially regulate the Wnt/β-catenin signaling pathway. Once hypertrophic maturation is complete, Runx2 is then subject to suppression by SOX9 [19]. Recent studies shed light on the involvement of epidermal growth factor receptor (EGFR) signaling in regulating EO in the developing growth plate, and its role in the remodeling of cartilage ECM into bone [20]. Similar studies show that tumor growth factor-α (TGF-α) inhibits articular chondrocyte phenotype through activating the Rho-kinase/ROCK and ERK signaling pathways (Figure 1B) [21].

So far, much work has been applied to decipher the intrinsic mechanisms initiating chondrogenic differentiation by MSCs. Chondrogenic differentiation leading to EO involves mesenchymal cells, pre-chondrocytes, differentiated chondrocytes, and hypertrophic chondrocytes [22] (Figure 1A). From one stage to another, various transcription factors, cytokines, and growth factors are recruited throughout the differentiation process [21]. Members of the Sox, SMAD, and BMP families [23,24] act as crucial regulators of chondrogenesis. Chondrogenesis is mediated by complex signaling pathways, including signaling of Wnt/β-catenin, fibroblast growth factor (FGF), Ihh, and parathyroid hormone-related peptide (PTHrP) pathways [25,26,27,28].

Different molecules and encoded proteins, especially transcription factors, promote the maturation of MSCs to mature chondrocyte cells [29,30,31]. Recent studies have demonstrated that transcriptional co-regulators significantly contribute to the epigenetic regulatory system [32].

Besides the genes and proteins, a number of studies have revealed that ncRNAs play vital roles in different biological events such as development, disease, and immunity [33,34]. Regarding epigenetic factors, it is obvious that miRNA and lncRNA play an important role in chondrogenesis and discuss in the following.

## 3. miRNA and Chondrocyte Differentiation

miRNAs are small (approximately 22 bp), single-stranded, and highly conserved molecules which regulate gene expression by pairing with the 3′ UTR of complementary mRNA sequences [35]. At least 60% of human genes contain target sites for miRNAs. mRNA targets are destroyed by a protein complex called RNA-induced silencing complexes at the post-transcriptional modification stage. Each miRNA may have several targets, which genetically regulate several biological processes such as chondrogenesis [35]. They can either reduce the target gene expression or recruit chromatin-modifying proteins, such as histone methyltransferases, which increase chromatin accessibility.

Different studies show that miRNA may facilitate chondrogenesis and regulate chondrocyte differentiation. For instance, in a study by Kobayashi et al., a chondrocyte-specific Dicer knockdown mouse model produced mice that did not express miRNAs in chondrocytes. This type of mice exhibited severe defects in skeletal development, as well as a reduction in chondrocyte proliferation and differentiation [36].

One of the more well-known chondrocyte-specific miRNA which has provided supporting knowledge of miRNAs involvement in cartilage development is miR-140 [37]. This miRNA is reported to be associated with SOX9 function and expression in mice [38]. miR-140 is synthesized from intron 16 of the WW Domain Containing E3 Ubiquitin Protein Ligase 2 (*Wwp2*) gene [39], which is a direct target gene of SOX9. Similar studies have unearthed the regulatory roles of miR-140 on the expression of other genes involved in chondrogenesis in human, including *SMAD3* and *RALA* [40]. Another study shows that SOX9 homozygous null zebrafish and human chondrocytes treated with SOX9 siRNA express high levels of miR-140, confirming a reciprocal relationship between miR-140 and SOX9 [38]. These studies showed that miRNAs, e.g., miR-140, can regulate cartilage development by targeting various targets.

Various studies indicate that miR-140 plays a role in chondrogenesis by indirectly mediating chondrocyte proliferation by inhibiting chondrocyte hypertrophy which maintains chondrocyte proliferation [41,42]. This process can be stimulated by targeting the expression of histone deacetylase 4 (HDAC4) with miR-140. This enzyme is expressed in proliferating chondrocytes and inhibits chondrocyte hypertrophy by decreasing the transcriptional activity of RUNX2 transcriptional activity [43]. miRNA-140 has been shown to target *HDAC4* in mouse cells. *HDAC4*-null mice display a skeletal phenotype, with early-onset chondrocyte hypertrophy and therefore premature ossification [43]. *HDAC4* gene is expressed in pre-hypertrophic chondrocytes and appears to regulate hypertrophy by inhibiting the action of RUNX2 [44].

Kenyon et al. showed that in addition to *SOX4*, other genes such as *KLF4*, *PTHLH*, and *WNT5A* can also be regulated by miR-140 [45] and this underlines the diverse targets of miRNAs. These genes have important roles in chondrogenesis. Further investigations suggested that miR-140–5p and miR-149 could mediate the development of osteoarthritis (OA), which is regulated by the expression of the gene *FUT1* [46]. The miR-140–5p/miR-149/FUT1 axis may serve as a predictive biomarker and a potential therapeutic target in OA treatment [46]. Novel studies show the important roles of miR-140 and miR-146 in inflammatory pathways related to OA. miR140 regulates the expression of toll-like receptor 4 (*TLR-4*), while miR-146 participates in the regulation of TLR-4 signaling through targeting *IRAK-1* and *TRAF-6* [47]. Furthermore, it has been proven that the overexpression of miR-140–5p hinders lipopolysaccharide-induced human intervertebral disc inflammation and degeneration by reducing or downregulating the TLR-4 [48]. It has been demonstrated that miR-146 is highly expressed in low-grade OA cartilage and that its expression is stimulated by IL-1β. It has been concluded that miR-146 may play a role in OA cartilage pathogenesis [49]. miR-140 also can change epigenetic mediations such as acetylation, resulting in the promotion of other downstream effects. For example, the cartilage-specific miRNA-140 targets *HDAC4* in mouse cells. *HDAC4*-null mice display a skeletal phenotype, with early-onset chondrocyte hypertrophy and therefore premature ossification [43]. *HDAC4* gene is expressed in pre-hypertrophic chondrocytes and appears to regulate hypertrophy by inhibiting the action of RUNX2 [44].

## 4. The Role of miRNAs in Cartilage Homeostasis

miRNAs can play a role in cartilage homeostasis. For instance, Li et al. showed that miR-140–5p may have a hand in regulating mandibular condylar cartilage homeostasis and potentially serve as a novel prognostic factor of TMJ-OA-like pathology in mice [50]. This study was comparable with research showing that the expression levels of miR-22, miR-140, and Bone Morphogenetic Proteins (BMP)-2 can show the alternation in synovial fluid of OA patients before and after arthroscopic debridement [51]. These findings demonstrated that miR-140 plays an essential but unknown role in the etiology of OA. Although deletion of miR-140 in mice results in slightly short stature characterized by shortened limbs, these mice exhibit an age-dependent OA phenotype [52]. This study can reveal the involvement of miR-140 in both physiological and pathological conditions by changing the cartilage tissues. Besides, miR-140 is essential for cartilage homeostasis in developing and adult stages and it is a candidate therapeutic target for OA treatment. For instance, miR-140 directly regulates *ADAMTS5* expression which is strongly associated with OA progression [52,53]. In addition, it was identified that the expression of *IGFBP5* (insulin growth factor binding protein-5) in human OA chondrocytes was significantly lower than that of controls and that *IGFBP5* is the direct target of miR-140 [54].

In cartilage homeostasis, TGF-β plays a dual role of regulating matrix catabolism by modifying gene expressions. Activated TGF-β promotes cartilage matrix anabolism or catabolism by different protein complexes formed by Smad family members. miR-146a targets Smad4 through both mRNA degradation and translational repression and affects the complexes formation. Thus, this miRNA can change matrix catabolism and anabolism. In a similar manner, miR-483 and miR-337 influence cartilage matrix homeostasis through BMP7 and TGF-β2R, respectively.

## 5. miRNAs Contribute to Chondrocyte Differentiation

In addition to miR-140, other miRNAs have been identified as regulators of chondrocyte differentiation (Table 1). For example, several studies have shown that the expression of miR-145 is blunted during chondrocyte differentiation of mouse mesenchymal cells [55,56]. This finding is recapitulated when miR-145 is inhibited, which triggers chondrocyte differentiation [57]. Conversely, overexpression of this miRNA attenuates chondrocyte differentiation by reducing the expression of various chondrocyte proteins such as COL2A1 and ACAN.

miRNAs including miR-495, miR-101, and miR-1247 directly regulate *SOX9* expression and inhibiting chondrocyte differentiation [93,94,95]. miR-574–3p is up-regulated by SOX9 which inhibits chondrocyte differentiation in a negative feedback-like manner [96]. Other studies showed that the overexpression of miR-675 enhances *COL2A1* expression in human chondrocytes which facilitate differentiation [97]. In a similar manner, *BMP2*-responsive miR-199a controls human chondrocyte differentiation through regulating *SMAD1* [98,99].

Xu et al. investigated the biological functions of miR-27b during hypertrophic differentiation of rat articular chondrocytes. According to in situ hybridization and immunohistochemistry, they reported a reduction of miR-27b expression in the hypertrophic zone of articular cartilage, but an increase of peroxisome proliferator-activated receptor-gamma 2 (*Pparγ2*) expression [100]. This study acknowledged that miR-27b regulates chondrocyte hypertrophy in part by targeting *Pparγ2*, and that the involvement of miRNA may have important therapeutic implications in cartilage diseases [100].

In 2018, it was shown that the PTHrP concentration gradient across the growth plate regulates the expression of various miRNAs such as miR-374–5p, miR-379–5p, and miR-503–5p, which in turn regulates chondrocyte proliferation and hypertrophic differentiation. The data support a model in which higher concentrations of PTHrP in proliferative zone upregulate the mentioned miRNAs, whereas, in the hypertrophic zone, lower concentrations of PTHrP cause downregulation of these miRNAs result in inhibiting the cell proliferation and promoting hypertrophic differentiation in rats [101].

Another important miRNA is miR-181b which was significantly down-regulated during chondrogenic differentiation of TGF-β3-stimulated limb mesenchymal cells, although it was up-regulated in osteoarthritic chondrocytes extracted from the cartilage of OA patients. Using either a mimetic or an inhibitor to alter the miR-181b levels in various cells such as chondroblasts and articular chondrocytes. The findings illustrated that the attenuation of miR-181b mitigated MMP-13 expression while promoting type II collagen expression. Forced expression of anti-miR-181b significantly decreased the cartilage destruction as resulted from DMM surgery in mice. In sum, these data suggest that miR-181b is a negative regulator of cartilage development, and inhibition of this miRNA could be implied as an effective therapeutic strategy for the cartilage-related disease [102].

Another research revealed that three types of miRNAs identified in chondrocytes are associated with age. This study showed that miR-199a-3p and miR-193b are involved in chondrocyte aging by regulating ACAN and type 2 collagen and that SOX9 and miRNA-320c cooperate in the juvenile phase of chondrocytes by regulating *ADAMTS5* [103]. This study also revealed the complex interaction network of miRNAs and their roles not only by a direct effect on target gene expression but also by promoting other mediator proteins to change the downstream genes expression.

Growth/differentiation factor-5 (GDF-5) plays an important role in initiating chondrocyte precursor cells condensation formation and also functions at two steps of skeletal development by two distinct mechanisms. Firstly, GDF-5 triggers the initial stages of chondrogenesis by promoting cell adhesion. This protein then enhances the size of the skeletal elements by promoting proliferation within the epiphyseal cartilage adjacent to its expression within the joint interzone [104]. Besides, various research identified *GDF-5* as the direct target of miR-21 during the regulation of chondrogenesis. Similar to the previous study, this study shows the important roles of miRNAs in chondrogenesis. Some studies show that miR-21 plays an important role in the pathogenesis of OA and can be deemed as a potential therapeutic target. Indeed, overexpression of miR-21 could attenuate the process of chondrogenesis [105].

Many studies demonstrated that depending on which miRNAs are present chondrogenesis can coincide with inflammation. For instance, it has been deducted that miR-320c expression is increased during the middle stage, but decreased during the late stage of hBMSC chondrogenic differentiation. On the other hand, *CDK6* expression decreased during the middle stage and increased during the late stage of hBMSC chondrogenic differentiation. In other words, *CDK6* and miR-320c not only co-regulate hBMSC chondrogenesis but also regulate the IL-1β-induced chondrocyte inflammation through the NF-kB signaling pathway. This study suggests that miR-320c and CDK6 inhibitors can be used to repress catabolism in human chondrocytes and affects chondrocytes homeostasis [106].

miR-1307-3p also inhibits the chondrogenic differentiation of human adipose-derived stromal/stem cells (hADSCs) by targeting Bone morphogenetic protein receptor type II *(BMPR2)* and its downstream signaling pathway, which may provide novel therapeutic clues for the treatment of cartilage injury [90]. BMPR2 is one of the important proteins playing roles in chondrogenesis. It is one of the essential members of the TGF- βfamily and is involved in paracrine signaling [107].

miR-892b has a key role in targeting the *KLF10*–*Ihh* axis as a regulator of hypertrophy in TGF-β-mediated chondrogenesis of hMSCs. This study provides a novel strategy for preventing hypertrophy in chondrogenesis from MSCs [108]. Interestingly, Kruppel-like factor 10 (KLF10) was demonstrated to be a target of miR-892b and also to act as a transcription factor which directly regulated Ihh expression by binding to its promoter. Similarly, other results give away that MiR-132–3p promotes rat MSCs chondrogenic differentiation, possibly mediated by targeting A disintegrin and metalloproteinase with thrombospondin motifs-5 (*ADAMTS-5*), which provided a new perspective on the chondrogenic differentiation and pathology of OA [109]. In fact, *ADAMTS-5* takes part in the processing of procollagens and cleavage of aggrecan, versican, brevican, and neurocan [110].

## 6. miRNAs, Environmental Factors, and Cartilage Development

Novel studies have disclosed that hormones or environmental factors can take a role in regulating miRNAs. For instance, Xu et al. showed that estrogen therapy reduced the degree of cartilaginous degeneration through upregulating the expression levels of miR-140–5p in cartilage and subchondral bone remodeling in an ovariectomized rat model of postmenopausal osteoarthritis [111]. Similarly, the administration of either miR-526b-3p or miR-590–5p successfully promoted the chondrogenic differentiation of human bone marrow stem cells (hBMSCs). Collectively, this study showed that the modification of hBMSCs using either melatonin or miRNA transduction could be an effective therapy for cartilage damage and degeneration. In sum, they revealed that melatonin promotes chondrogenic differentiation of human hBMSCs via upregulation of miR-526b-3p and miR-590–5p. Mechanistically, the elevated miR-526b-3p and miR-590–5p exert their effects via the Smad family. In other words, they enhanced SMAD1 phosphorylation by targeting *SMAD7* [112].

## 7. Smad Family and Cartilage Development

As discussed, Smad proteins play an important role in chondrogenesis. They are intracellular molecules that mediate the canonical signaling cascade of TGF-β superfamily [113]. However, the distinct mechanisms utilized by these proteins to initiate nuclear events and their interactions with cytoplasmic proteins are still widely unknown. In a study, Xu et al. uncovered that the forced expression of *SMAD6* significantly promoted chondrogenic differentiation. However, the SMAD6-induced chondrogenic differentiation could be reversed by miR-134 mimetics. In conclusion, miR-134 acts as a negative regulator during chondrogenic differentiation of hBMSCs through interacting with *SMAD6* [114].

## 8. Cartilage Development and Epigenetic Modulations

miRNAs can influence various epigenetic factors such as histone deacetylase family. Therefore, they exert their functions indirectly in addition to their direct actions. For example, in 2019, Zhang et al. reported that miR-193b-5p downregulated the expression of *HDAC7, MMP3*, and *MMP13*. This miRNA regulates chondrocytes metabolism by directly targeting histone deacetylase 7 in interleukin-1β-induced OA [115]. Indeed, chondrocytes are able to release adenosine upon a variety of physiological stimuli. These adenosine levels, in either the extracellular or endogenous setting, play an important role in regulating cartilage damage during inflammatory processes. If these levels drop, the expression of glycosaminoglycans, matrix metalloproteinases (particularly *MMP3* and *MMP13*), and nitric oxide will increase to promote the chondrogenic differentiation [116]. In 2018, it was ascertained that miR-95–5p inhibits HDAC2/8 expression and promotes cartilage matrix expression. The results suggest that AC-miR-95–5p-Exos regulate cartilage development and homeostasis by directly targeting HDAC2/8. Thus, this miRNA may act as an HDAC2/8 inhibitor and exhibit potential as a disease-modifying OA drug [117].

## 9. miRNAs and Their Protective Roles in Cartilage

Some of the identified miRNAs have a hand in protection in cartilage development. For example, the overexpression of miR-455–3p inhibits the degeneration process during chondrogenic differentiation, while deletion of this miRNA in mice accelerates cartilage degeneration. Indeed, epigenome-wide association study verified that the forced expression of this miRNA regulates DNA methylation of cartilage-specific genes. Consequently, Gene Ontology analysis revealed that the PI3K-Akt signaling pathway was mostly hypomethylated. Therefore, miR-455–3p can regulate hMSC chondrogenic differentiation by influencing on DNA methylation [118]. miR-181a also might be necessary for chondrogenesis, as demonstrated in pig peripheral blood MSCs. However, the distinct mechanism by which miR-181a regulates the chondrogenesis still needs further investigations [119].

In 2019, Huang et al. showed that miR-204 and miR-211 are essential in maintaining healthy homeostasis of mesenchymal progenitor cells (MPCs) to counteract OA pathogenesis in mouse models [120]. Any perturbation, leading to the loss-of-function of miR-204 or -211 in MPCs, results in aberrant accumulation of Runx2, abnormally increases proliferation and osteogenic differentiation of MPCs, induces matrix-degrading proteases in articular chondrocytes, and importantly stimulates articular cartilage destruction. In a nutshell, it generates comprehensive OA phenotype [120]. This study affirms the important protective role of miR-204 and miR-211 in cartilage development.

Although many miRNAs have been identified as regulators of chondrocyte differentiation, their biological significance remains unclear.

## 10. Role of lncRNAs in Chondrogenesis

lncRNAs are substantial regulators of transcriptional and post-transcriptional modification. Contrary to miRNAs, most lncRNAs (around 81%) are poorly conserved, though some lncRNAs have been discovered to be conserved among species for more than 300 million years. Some lncRNAs are mainly located in the cytosol, where they target mRNAs and downregulate protein translation. Interestingly, they may also act as decoys for miRNAs, thus preventing the inhibitory effect of the binding of miRNAs to their target mRNAs [121].

It has been proven that lncRNAs play an important role in the development and progression of chondrogenesis [122], as summarized in Table 2. Specifically, lncRNA-HOXA transcript induced by TGF-β (HIT) not only is expressed in non-small cell lung cancer [123,124] but also exists in E11 mouse embryos; lncRNA-HIT is correlated with p100 protein and CREB-binding protein (CBP) to make a regulatory complex called lncRNA-HIT-p100/CBP during chondrogenic differentiation [125]. Any perturbations for *lncRNA-HIT* expression result in no formation of cartilage. Thus, lncRNA-HIT can be deemed as an indispensable factor for mouse chondrogenic differentiation in the limb mesenchyme [126].

Both ADSCs and hBMSCs can differentiate into many cell lineages. In 2019, Pang et al. who focused on the chondrocytic differentiation potential of ADSCs to decipher the mechanisms with the understanding that the lncRNA H19 partakes in this process. Actually, the overexpression of H19 in ADSCs induced differentiation towards chondrocytes. This lncRNA is amply expressed during embryonic development, but downregulated after birth, implying its regulatory role in determining cell fate. They showed that H19 exerts its regulatory function during human cartilage differentiation of ADSCs through competing for miRNA regulation of STAT2 [136]. According to another study, during the processes of chondrogenic differentiation of MSC, UCA1, miR-145–5p, or miR-124–3p are overexpressed. UCA1 is a long noncoding RNA which substantially improves chondrogenesis of MSCs. This lncRNA down-regulates the expression of miR-145–5p and miR-124–3p, which attenuate the chondrogenic differentiation of MSCs. UCA1 consequentially stimulates TGF-β pathway members (SMAD5 and SMAD4), which is targeted by miRNA-145–5p and miRNA-124–3p [134]. These data pave the way to better understand the lncRNAs role in chondrogenesis.

Other studies show the potential roles of lncRNAs’ partaking in in vitro or artificially engineered cartilage. For instance, in 2019, Huynh et al. identified a novel lncRNA, Glycosaminoglycan Regulatory ASsociated Long Non-coDing RNA (GRASLND) as a regulator of MSCs chondrogenesis. This lncRNA is a primate-specific lncRNA, is upregulated during MSC chondrogenesis, and appears to act directly downstream of SOX9. Using the established models, they showed that the silencing of GRASLND resulted in lower accumulation of cartilage-like ECM, while the overexpression of this lncRNA, either via transgene ectopic expression or by endogenous activation via CRISPR, significantly enhanced cartilage matrix production. This novel lncRNA acts to inhibit interferon-gamma (IFN-γ) by binding to eukaryotic initiation factor-2 kinase (EIF2AK2). In a nutshell, they elucidated that GRASLND enhances chondrogenesis via suppression of interferon type II signaling pathway [127].

During hBMSCs chondrogenic differentiation, the altered expression of the lncRNAs called, *ZBED3-AS1* gene was elucidated [137]. Then, another study confirmed that this lncRNA’s expression fluctuates during the chondrogenic differentiation of human synovial fluid-derived MSCs (SFMSCs) [132]. Further complementary studies have verified that the forced expression of this lncRNA enhances the chondrogenic potential by regulating the Wnt/β-catenin signaling pathway via the upregulation of zbed3 [138].

Recently, it has been debated that increased expression of lncRNA DANCR (differentiation antagonizing non-protein coding RNA) stimulated by SOX4, enhances proliferation and chondrogenesis of synovium-derived mesenchymal stem cells. This upregulated lncRNA leads to activating the expression of various genes such as *MYC*, *SMAD3*, and *STAT3* by direct interaction with their mRNAs [139]. As well as, they demonstrated that the crosstalk among DANCR, miR-1305, and Smad4 significantly contributes an essential role in SMSC proliferation and differentiation [140]. This study shows that lncRNAs exert their functions directly or indirectly. Other important lncRNAs are summarized in Table 3.

## 11. Other ncRNAs and Cartilage Development

Other members of ncRNAs including long intergenic non-coding RNAs (lincRNAs), small nucleolar RNAs (snoRNAs), and small nuclear RNA (snRNA) are also partaking important roles in cartilage development. LincRNAs are a subgroup of lncRNAs that are >200 bp long and are emerging as important regulators of diverse biological processes. Pearson et al. identified that the lincRNA p50-associated cyclooxygenase 2–extragenic RNA (PACER) and CILinc01 and CILinc02 as novel chondrocyte inflammation-associated lincRNAs were differentially expressed in both knee and hip OA cartilage compared to non-OA cartilage. Indeed, this indicates that the lincRNAs do a protective role in preventing inflammation-mediated cartilage degeneration [156]. In 2019, Rai et al. compared the transcriptome of articular cartilage from knees with meniscus tears to knees with end-stage OA and demonstrated that cartilage derived from the aforementioned samples differentially express eight lincRNAs including PWRN1, FAM157A, TTTY12, C9orf163, C22orf34, MGC45922, C10orf91, and MEG3 [157]. Additionally, they showed a difference in the expression pattern of the derived chondrocytes regarding snoRNAs.

SnoRNAs do enzymatic modifications of other RNAs, e.g., ribosomal RNAs, by forming ribonucleoprotein complexes with enzymes [158]. There are two fundamental classes of snoRNAs: the C/D box snoRNAs (SNORDs) and H/ACA box snoRNAs (SNORAs), which are correlated with methylation and pseudouridylation of ribosomal and other RNAs [158]. There is a paucity of data showing the roles of snoRNAs in cartilage development. Controlling snoRNA expression may play a vital role in the regulation of chondrocytes [159]. RNase MRP is another member of snoRNAs which is identified as a regulator of chondrocyte hypertrophy [160]. Specific alternations in snoRNAs abundance in joint aging have been shown; SNORD88 and SNORD38 were respectively decreased and increased in old equine cartilage in human-derived samples [161]. Whilst other studies demonstrated that snoRNAs U38 and U48 were significantly increased in sera of patients developing cartilage damage [162], however the exact function of these RNAs in cartilage is unclear.

In eukaryotic cells, small nuclear RNAs (snRNAs) play a critical role in splicing speckles and Cajal bodies of the cell nucleus. Very little information is available about the contribution of such ncRNAs to cartilage development. It has been demonstrated that miR-1246, which is upregulated at a later stage in chondrogenesis, is a pseudo-miRNA nuclear fragment of the longer U2 small nuclear RNA [163,164].

## 12. ncRNAs and Cartilage Pertinent Diseases

Numerous diseases affecting cartilage are called chondropathies and can be divided into the various subgroups, but in this review, we focused on some of the well-known diseases involving achondroplasia, OA, spinal disc herniation, human cartilage-hair hypoplasia, age-related degeneration of articular cartilage, and brachydactyly type E.

To begin with, achondroplasia is the most common form of short-limb dwarfism in human and affects more than 250,000 individuals worldwide [165]. This disease is caused by mutations in fibroblast growth factor receptor 3 (FGFR3). These mutations affect the cartilaginous growth plate [166]. Although it has not been detected any ncRNAs related to achondroplasia, there is a universal attempt to treat the patients with FGF-R mutations by using ncRNAs [167].

It was shown that in patients affected by cartilage-hair hypoplasia (CHH), mutations in RMRP which is the RNA component of the ribonucleoprotein complex RNase MRP, upregulation of several cytokines and cell cycle regulatory genes are obvious [168]. CHH is a disease characterized by disproportionate short stature, fine and sparse hair, deficient cellular immunity and a predisposition to malignancy. This study suggested that alteration of ribosomal processing in CHH is correlated with changed cytokine signaling and cell cycle progression in terminally differentiating cells in chondrocytic cell lineages.

There are reports on the loss-of-function mutations of miR-140 in different abnormalities [37,53], however, in 2019, Grigelioniene et al. detected gain of a new function of miR-140 in human skeletal dysplasia [169]. They demonstrated a delayed maturation of the epiphyses and growth plate chondrocytes in addition to absent or severely impaired cartilage mineralization in mice. These features were highly consistent with the skeletal dysplasia features of the patients, including delayed secondary ossification, mild platyspondyly, small epiphyses, and scaphocephaly. This study provides insight into neomorphic actions of emerging and/or mutant miRNAs.

Most information we know about the ncRNAs comes from OA. Multifarious studies have been applied to show the alternation of miRNA expression during OA [170]. For example, it was identified that miR-22 directly targets *PPAR* and *BMP7* in association with OA [171]. Similarly, as we discussed, miR-146a represses IL-1-regulated gene expression had been identified by Diaz-Prado et al. After that, a number of miRNAs showing differential expression including miR-483 and miR149 added to the list of ncRNAs with impacts on OA [172].

In 2015, it was shown that human *MEG3* was significantly downregulated in OA patients in comparison with the normal cartilage samples [173]. Other studies scrutinized that *MEG3* knockdown might lead to the progression of OA through the miR-16/SMAD7 axis; absolutely, MEG3 was down-regulated, and miR-16 was up-regulated in cartilage tissues of rat OA model [174]. Further investigations showed that *MEG3* levels are inversely associated with VEGF levels; hence, *MEG3* may be involved in OA development through the regulation of angiogenesis [173]. Similarly, another group found that the introduction of miR-21 into the cartilage of OA mice significantly stimulated cartilage destruction. Taken together, the results showed that GAS5 contributes to the pathogenesis of OA by acting as a negative regulator of miR-21 and thereby regulating cell survival [175].

Using novel techniques such as microarray analysis, Xing et al. identified 121 lncRNAs that were either up- or down-regulated in OA tissue compared with normal tissue, 73 of those lncRNAs were upregulated, and 48 were downregulated compared with normal cartilage. Approximately 21 of the differently expressed lncRNAs were up-regulated by almost two-fold. Expression of 6 lncRNAs, including HOTAIR, GAS5, PMS2L2, RP11–445H22.4, H19, and CTD-2574D22.4, were up-regulated in OA in comparison with the normal tissue as validated by RT-PCR after microarray analysis. Expression of mRNA for *MMP-9*, *MMP-13*, *BMP*-2, and *ADAMTS5* in OA was significantly greater than in normal cartilage. However, the expression of mRNA for *COL2A1* was lower in OA than in normal cartilage [176]. This research confirmed the necessity of using novel techniques to detect the expression of lncRNAs in chondropathy (Table 3).

It was revealed that the lncRNA RP11–445H22.4, miR-301a, and CXCR4 axis played an important role in cartilage ATDC5 cells and provided a theoretical basis of lncRNA RP11–445H22.4 in a mouse model of OA [177]. Also, the present study suggests that XIST promotes MMP-13 and ADAMTS5 expression, indicating ECM degradation, by functioning as a competing endogenous (ceRNA) of miR-1277-5p in OA [178]. This study proposed a novel potential target with a new working mechanism in OA treatment [178]. HOTAIR/miR-17–5p/FUT2 axis was also indicated that contributed to OA progression in human via the Wnt/β-catenin pathway, which might provide novel insights into the function of lncRNA-driven OA [179].

Advances in novel sequencing technologies such as next-generation sequencing and particularly whole-genome sequencing has helped scientists to find out novel ncRNAs in OA. For instance, a study applied in 2019, based on 19 samples from OA patients detected around 580 significantly dysregulated lncRNAs in addition to the differently expressed *TUCPs* and mRNAs. They finally, confirmed four differently expressed lncRNAs (*SNHG5*, *ZFAS1*, *GAS5*, and *DANCR*) involved in OA cartilage by protein-protein interactions network and ceRNA regulatory network [180]. The experiment applied by Ali et al. showed that next generation sequencing is a useful approach for identifying known and novel circulating miRNA signature in cartilage pertinent disease [181]. These results could expand our horizons toward OA mechanism. We summarized some of the important studies regarding cartilage diseases in Table 3.

## 13. Therapeutics of ncRNAs on the Cartilage Diseases

It has been suggested that ncRNAs can be used for diagnosis and prognosis of cartilage-pertinent diseases, and might also be exploited as potential therapeutic agents to stop and even reverse the disease progression. Some studies asserted that inhibition of GAS5 will suppress cell apoptosis and probably can be used as a therapeutic agent [182]. It has been detected that exosomes derived from miR-140–5p-overexpressing human synovial MSCs promote cartilage tissue regeneration and prevent OA of the knee in a rat model [183]. Knockdown of HOTAIR has also been proven that led to growth inhibition of human chondrosarcoma cells via G0/G1 arrest and apoptosis in vitro and in vivo. This study is one of the first investigations paying to lncRNAs and their utilization in therapeutic application [184]. In the same way, in 2019, it was demonstrated that the downregulation of LOC101928134 suppresses the synovial hyperplasia and cartilage destruction of OA rats via activation of JAK/STAT signaling pathway by upregulating IFNA1, providing a new candidate for the treatment of OA [185].

More recently, Peng et al. showed that overexpression of insulin-like growth factor 1 (IGF-1) and miR-140 in articular chondrocytes improves cartilage repair [186]. This study can attract the attention to use miR-140 as a therapeutic treatment for degenerative diseases such as OA. In another similar study, they report that the repair of full-thickness cartilage defects in a rabbit model can be triggered by overexpression of IL-1Ra and miR-140 [186,187].

Circular RNAs or circRNAs, categorized as ncRNAs, could also be used as a potential target in the treatment of cartilage diseases. For example, circRNA-CER regulates MMP13 expression by acting as a ceRNA and participates in the process of chondrocyte ECM degradation [188]. In fact, ceRNA regulates other RNA transcripts by competing for shared miRNAs and can derepress all target genes of the respective miRNA family [188]. Besides, hsa_circ_0045714 was introduced as a circRNA which can regulate synthesis in addition to proliferation and apoptosis of chondrocytes by promoting the expression of miR-193b target gene *IGF1R*. This study provides new insight on the applications of circRNA in OA and other cartilage pertinent diseases [189].

The above studies can show the potential efficacy of ncRNAs as the novel agents for therapeutic purposes. However, the amount of current knowledge on ncRNAs and their distinct functions that affect the progression of cartilage pertinent diseases is still limited. Taken together, although ncRNAs able to serve as noninvasive biomarkers or therapeutic agents, their broad utilization is restricted in cartilage and bone diseases. Actually, accumulating evidence suggests that ncRNAs participate in a variety of diseases and many of them affect crucial signaling pathways which can result in several diseases [190]. The potential use of ncRNAs for therapies is tremendous, however, before being capable of using these new therapeutic options commonly, more functional and structural studies are necessary, particularly in animal models, to better understand the biology of ncRNAs to mitigate the unwanted side-effects.

Furthermore, it has been proven that RNA-based therapeutics is a tough challenge and have several limitations such as rapid degradation, abnormal innate immune responses, difficult dosage determination, and off-targeting [191]. As discussed, miRNAs often target several mRNAs related to different genes, e.g., in Figure 1, we showed that miR-140 has various targets. Also, another challenge for bridging the gap between lab and clinic is the side effects of ncRNAs related therapeutics. In the last decade, many techniques and tools promoted scientists to determine the target of regulatory RNAs, especially miRNAs, in the human genome. It indicates that miRNAs are nonspecific for a particular gene, and suppressing them can thus cause many side effects [192,193].

## 14. Conclusions

Epigenetic regulation attracts attention toward the important biological process and is well established in many research fields; however, the nature of epigenetic factors and their roles in skeletal systems are shrouded in mystery. Because of cutting-edge technologies such as whole-genome sequencing and remarkable progress in epigenetic studies, many epigenetic factors have been identified which have the potential to be a therapeutic target for many diseases and for regenerative medicine purposes. It should be noted that miRNAs are a critical regulator for gene expression and exponents a promising therapeutic target pertinent to cartilage diseases. More knowledge about epigenetic regulation of chondrocyte differentiation would contribute to the development of treatments and therapies for cartilage diseases such as OA.

We discussed ncRNAs and their roles in cartilage development and cartilage diseases. Although a large number of ncRNAs have been determined related to cartilage development, only a small portion of their functions has been elucidated. As we debated, there is a network on the interaction among various epigenetic factors in cartilage development or chondrogenesis and there is no possibility that a single ncRNA could be responsible for the cartilage destruction or clinical symptoms. Therefore, it seems that it is important to detect further ncRNAs which have hand in the development of cartilage and its diseases. Maybe one of the well-known resources to study the ncRNAs is using cartilage tumors. This source provides valuable information about the biological aspects of ncRNAs and their involvement in development and diseases. We are optimistic that future endeavors will reveal the roles of ncRNAs in skeletal biology.

## Figures and Tables

**Figure 1 ijms-20-04475-f001:**
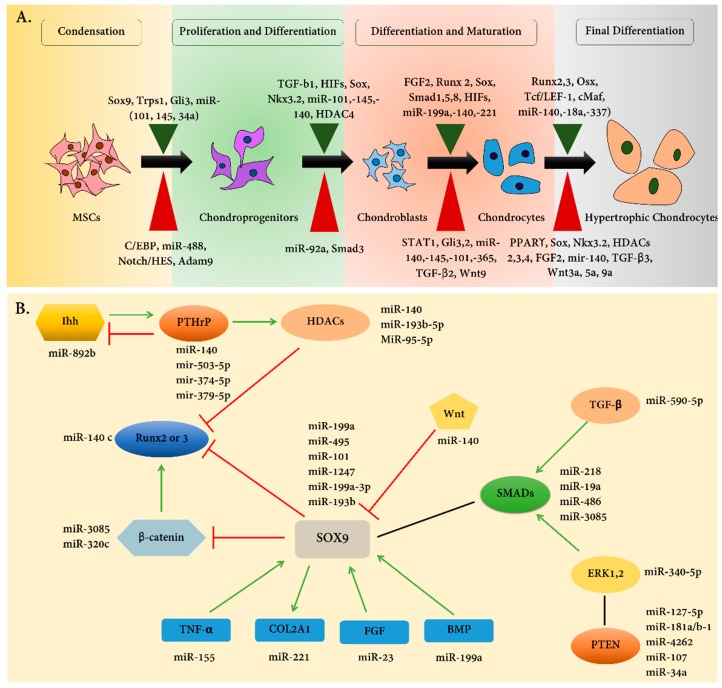
(**A**) Main steps of chondrogenesis and the signaling factors involved in chondrocyte development. MSCs can be differentiated to chondroprogenitor cells which are initially loosely arranged with elongated mesenchymal cell morphology. Precartilaginous nodules resulting from chondrogenic cells extensive proliferation and migration can be regulated by major transcription and growth factors. The positive regulators are shown on the top panel in green, whereas the inhibitory regulators are depicted in red at the bottom panel. After that the differentiation process has transpired, as a result of cytoskeletal rearrangement, cells gain their round morphology and obtain a gene expression pattern required for cartilage-specific extracellular matrix production and maintenance. MSC: mesenchymal stem cells, miR: microRNA. (**B**) Different combinations of transcription factors regulating chondrogenesis. We summarized some of the well-known miRNAs and their targets. Red lines show the inhibitory interactions, green lines reveal the activating combination, and black lines just show the interaction between two protein families beside the inhibitory or activating manners. The early steps of chondrogenesis are crucially dependent upon SRY-box containing gene (Sox)-family transcription factors, which have great interactions with other protein factors. By contrast, Runx2 and Runx3 regulate the process of chondrocyte hypertrophy. The interacting miRNAs are shown in this figure. Some of them, miR-140, have various targets, while others have specific targets.

**Figure 2 ijms-20-04475-f002:**
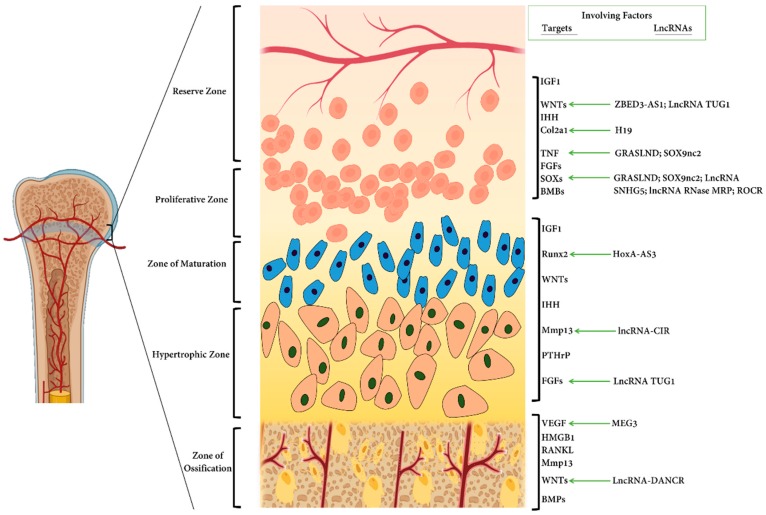
The growth plate can be divided into different distinct areas, each populated with cartilage cells (chondrocytes) displaying characteristic behaviors. Nearest the epiphyses is called a reserve or germinal zone involving stem cells. This region functions as a reserve of precursor cells for the proliferating chondrocytes in the columns. The second zone is a zone of proliferating chondrocytes. In this section, various genetic factors such as Indian hedgehog (Ihh) family are acting and the cells flatten and divide, laying down a cartilage ECM that will later play as a scaffold for bone formation. In the hypertrophic zone, the chondrocytes are still organized in columns, but they become ~5–12 times bigger and the amount of ECM increases, and its composition also altered. At the bottom end of the growth plate, the chondrocytes undergo apoptosis which triggers blood vessel invasion and are replaced by bone. What determines the size of these regions is still unclear. Marker/regulator gene expression in these different zones of the growth plate and the perichondrium are listed. The important long non-coding RNAs are summarized by showing their direct targets. Also, the important lncRNAs playing a major role in regulating chondrogenesis are shown. Indeed, many factors contributing to chondrogenesis are a target for their specific lncRNAs and some of the protein families, for instance Wnt family proteins, play important roles in various stages.

**Table 1 ijms-20-04475-t001:** Important microRNAs involving in chondrogenesis and chondrocyte development.

miRNA	Abnormal Expression (↑ Or ↓)	Target	Cell Type	Year of Publication	References
miR-140	-	Targets RALA, histone deacetylase 4, aggrecanases and syndecan 4	Mouse chondrocytes	-	[40,41,58]
miR-3085	↑	Regulates negatively transforming growth factor β1/Smad, IL-1β/NFκB, and Wnt3a/β-catenin signaling pathways	Mouse and rat chondrocytes	2018	[59]
miR-125b-5p	-	Targets TRAF6/MAPKs/NF-κB pathway	Human chondrocytes	2019	[60]
miR-138	↑	Downregulates RhoC and the Actin Cytoskeleton	Human chondrocytes	2018	[61]
miR-218	↑	Suppresses ALP, BSP, collagen type II alpha 1, OCN, and osteopontin. Promotes SOX9, COL2A1, ACAN, GAG, and COMP	Human synovium-derived MSCs	2019	[62]
miR-340–5p	-	Inhibits the ERK signaling pathway via the FMOD gene	Mouse chondrocytes	2018	[63]
miR-186	↓	Interacts with SPP1 and regulates PI3K–AKT pathway	Mouse chondrocytes	2018	[64]
miR--27b-3p	↓	Inhibits HIPK2 expression	Human chondrocytes	2018	[65]
miR-193b-3p	↑	Inhibits matrix metalloproteinase-19	Human chondrocytes	2018	[66]
miR-24	↑	Targets C-myc on apoptosis and cytokine expressions in chondrocytes of osteoarthritis rats via MAPK signaling pathway	Rat chondrocytes	2017	[67]
miR-9–5	↑	Inhibits apoptosis of chondrocytes through downregulating Tnc in mice with osteoarthritis	Mouse chondrocytes	2019	[68]
miR-27a	↑	Inhibits PLK2	Knee arthritis rat model	2019	[69]
miR-495	↑	Targets AKT1	Rat	2019	[70]
miR-93	↑	Targets TLR4/NF-κB signaling pathway	Mouse cartilage cells	2018	[71]
miR-31	↑	Targets C-X-C motif chemokine ligand 12	Human chondrocytes	2019	[72]
miR-590–5p	↑	Targets TGF-β1 to promote chondrocyte apoptosis and autophagy in response to mechanical pressure injury	Human chondrocytes	2018	[73]
miR-206	↓	Influences apoptosis and autophagy of articular chondrocytes via modulating the PI3K/protein kinase B-mTOR pathway by targeting insulin-like growth factor-1	Rat chondrocytes	2018	[74]
miR-23	-	Inhibits articular cartilage damage recovery by regulating MSCs differentiation to chondrocytes via reducing fibroblast growth factor 2	Rat MSCs	2019	[75]
miR-19a	-	Promotes cell viability and migration of chondrocytes via up-regulating *SOX9* through NF-κB pathway	Human synovium-derived chondrocytes	2018	[76]
miR-107	-	Regulates autophagy and apoptosis of OA chondrocytes by targeting *TRAF3*	Rat cartilage cells	2019	[77]
miR-98	-	Targets the 3́ untranslated region of Bcl-2	Human chondrocytes	2018	[78]
miR-615–3p	↑	Increases the expressions of inflammatory cytokines and inhibiting chondrogenic differentiation of hBMSCs.	Human bone marrow stem cells (hBMSCs)	2019	[79]
miR- 29b	↑	Regulates chondrogenesis homeostasis and enhance hypertrophic phenotype.	Murine MSCs	2019	[80]
miR-4784	↑	Promotes the expression of *Col2a1* and inhibits the *MMP*-*3* expression in chondrocytes.	Human cartilage cells	2018	
miR-320c	↑	Regulates the expression of β-catenin by directly targeting 3′UTR of β-catenin mRNA and decreasing the relative transcriptional activity of the β-catenin/TCF complex	human adipose-derived stromal/stem cells (hADSCs)	2019	[81]
miR-17-5p	↓	Induces autophagy mainly through suppressing the expression of p62	Mouse model of OA	2018	[82]
miR-181a/b-1	↑	Modulates PTEN/PI3K/AKT signaling and mitochondrial metabolism	Human chondrocytes	2019	[83]
miR-4262	-	Regulates chondrocyte viability, apoptosis, and autophagy by targeting *SIRT1* and activating PI3K/AKT/mTOR signaling pathway	Rat model of OA	2018	[84]
miR-221	↓	Increases the expression of typical chondrogenic markers including *COL2A1*, *ACAN,* and *SOX9*	Human cartilage cells derived from intervertebral disk	2018	[85]
miR-107	-	Modulates chondrocyte proliferation, apoptosis, and ECM synthesis by targeting *PTEN*	Human cartilage cells	2019	[86]
miR-34a	↑	Enhances chondrocyte apoptosis, senescence and facilitates the development of OA by targeting *DLL1* and regulating PI3K/AKT pathway	Rat chondrocytes	2019	[87]
miR-373	↓	Regulates inflammatory cytokine-mediated chondrocyte proliferation in OA by targeting the P2X7 receptor	Human chondrocytes	2018	[88]
miR-155	-	It is a sustainable factor for intervertebral disk and suppresses the expression of catabolic genes induced by TNF-α and IL-1β by targeting C/EBPβ in rat NP cells	Mouse cartilage cells derived from intervertebral disk	2018	[89]
miR-486	↑	Inhibits chondrocyte proliferation and migration by suppressing SMAD2	Human cartilage cells	2018	[90,91]
miR-127–5p	-	MALAT1/miR-127–5p regulates osteopontin-mediated proliferation of human chondrocytes through PI3K/Akt Pathway	Human cartilage cells	2018	[92]

**Table 2 ijms-20-04475-t002:** Summary of lncRNAs and their roles in chondrogenesis.

lncRNAs	Downstream Targets	Cell Type	Effects	Reference
GRASLND	*SOX9*	MSC	It has a protective effect in engineered cartilage against interferon type II across different sources of chondroprogenitor cells.	[127]
lncRNA TUG1	Notch and NF-κB pathways	murine chondrogenic ATDC5 cells	It suppresses Notch and NF-κB pathways.	[128]
SOX9nc2	SOX9 and TGFβ	OA cartilage	Depletion of the SOX9nc2 transcript, by RNAi, prevents chondrogenesis and concomitant induction of SOX9 expression.	[129]
H19	COL2A1	OA-affected cartilage	This lncRNA stimulates chondrocyte anabolism.	[130]
DANCR	Smad 4, miR-1305 STAT3, Smad3, myc	SMSCs	SOX4 could directly bind to the promoter of DANCR lncRNA and increases its expression and knockdown of DANCR could reverse these effects.	[131]
HIT	p100, CBP	mouse embryos	LncRNA-HIT is essential for chondrogenic differentiation in the limb mesenchyme.	[125]
ZBED3-AS1	zbed3	SFMSCs	It promotes chondrogenesis and could directly increase zbed3 expression.	[132]
ROCR	SOX9	human BMSCs	SOX9 induction is significantly ablated in the absence of ROCR. Thus, ROCR contributes to SOX9 expression and chondrogenic differentiation.	[133]
UCA1	miRNA-145–5p/miRNA-124–3p	human BMSCs	It promotes chondrogenic differentiation of human BMSCs via miRNA-145–5p/SMAD5 and miRNA-124–3p/SMAD4 axis.	[134]
PMS2L2	miR-203.	ATDC5 chondrocytes	PMS2L2 has a protective role in LPS-induced inflammatory injury in chondrocytes.	[135]

**Table 3 ijms-20-04475-t003:** The novel lncRNAs contribute to cartilage diseases.

Disease	Non-coding RNA	Abnormal Expression	Targets	Effects	Reference
Osteoarthritis	lncRNA H19		-	Regulates the balance between ECM anabolism and regeneration	[126]
	lncRNA HOTAIR	↑	matrix metalloproteinase (MMP)-1, MMP3, MMP9	Contributes to IL-1β-induced MMP overexpression and chondrocytes apoptosis in temporomandibular joint osteoarthritis.	[141]
	lncRNA PVT1	↓	miR-488–3p	Promotes apoptosis of OA and normal chondrocytes through miR-488–3p	[142]
	lncRNA-CIR	-	Bim and miR-130a	lncRNA-CIR/miR-130a/Bim axis is involved in oxidative stress-related apoptosis of chondrocytes in OA.	[143]
	lncRNA PACER	↓	down-regulates lncRNA HOTAIR.	Regulates chondrocyte apoptosis	[144]
	LncRNA CASC2	↑	IL-17	Participates in the regulation of IL-17 expression and chondrocyte proliferation and apoptosis.	[145]
	lncRNA Nespas	↑	ACSL6	suppresses miRs targeting ACSL6 and subsequent ACSL6 upregulation.	[146]
	LncRNA SNHG5	-	SOX2	LncRNA SNHG5/miR-26a/SOX2 signal axis enhances proliferation of chondrocyte in osteoarthritis	[147]
	lncRNA DILC	↓	IL-6	Regulates IL-6 expression in chondrocytes.	[148]
	LncRNA ANCR	↑	TGF-β1	ANCR might participate in OA by downregulating TGF-β1 and promote the proliferation of chondrocytes.	[149]
	lncRNA XIST	-	CXCR4 and MAPK signaling	lncRNA XIST can promote the proliferation of OA chondrocytes and promote apoptosis through the miR-211/CXCR4 axis	[150]
	LncRNA-p21	↑	miR-451	Negatively regulates the expression of miR-451 and promotes the apoptosis of chondrocytes in OA by acting as a sponge for miR-451	[90]
	LncRNA FOXD2-AS1	↑	miR-27a-3p	miR-27a-3p mimics could abolish the effects of FOXD2-AS1 overexpression on cell proliferation, inflammation, and ECM degradation in chondrocytes	[151]
Spinal disc herniation	lncRNAs TCONS	-	-	These lncRNAs in the spinal cord of rats in a radicular pain model of LDH have different expression patterns and their roles in unknown.	[152]
Human cartilage-hair hypoplasia	lncRNA RNase MRP	-	PTCH2 and SOX4	have gene-silencing activity relevant to human cartilage–hair hypoplasia	[153]
Age-related degeneration of articular cartilage	CRNDE and LINC00152 as the key lncRNAs	-	-	Are involved in the process of age-related degeneration of articular cartilage	[154]
Brachydactyly type E	DA125942	↑	PTHLH and SOX9	Upregulation of DA125942 downregulates SOX9.	[155]

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
