# Peer review of "Non-Coding RNAs in Cartilage Development: An Updated Review"

_ijms, 2019, doi:10.3390/ijms20184475_

Round 1
Reviewer 1 Report
Razmara et al “Non-Coding RNAs in Cartilage Development: An Updated Review”. An article reviewing cartilage development and its regulation by non-coding RNAs.
General comment: An ambitious manuscript, which provides a quick overview of endochondral bone formation and cites more than 100 references. Overall structure is good, however several details to be improved. The flow and the logic of it is not always easy to understand and the manuscript looks like a collection of spread facts/data summaries which seem to be difficult to be put together as pieces of one puzzle. The logics and flow of th presented facts need some revision.
Abbreviations are used without first time spelling out. English needs some minor revision.
Authors emphasize that microRNAs could be used as therapy for osteoarthritis, or cartilage disease, however, do not inform about pitfalls of this eventual therapy. I think that overall they are too optimistic considering that microRNAs regulate many more genes, than it was reported by earlier studies. Current techniques such as trancriptome studies have demonstrated that many more genes are involved than previously thought, indicating that this therapy may lead to unexpected side effects.
Overall through the review article the authors do not separate on data obtained using human cells, experimental animals, or human subjects. This is an important issue because there are important differences in experimental animals vs humans in the different processes involved in endochondral bone formation.
Further, I will now not separate to different sections and subsections of the manuscript, but will refer to row numbers.
Row 72. Chondrodysplasia generally is not so commonly used term anymore, I recommend changing it to skeletal dysplasia.
Rows 117-118. “Different genes and transcription factors promote the maturation of MSCs to mature chondrocyte cells (missing references). “ Different genes code for transcription factors and other important proteins in chondrogenesis thus I think it is important to use “different molecules promote”…
“Recent studies have demonstrated that transcriptional co-regulators significantly contribute to the epigenetic regulatory system (missing references).
Rows 147 – 148. Various other reports indicate that miR-140 plays a role in chondrogenesis by indirectly mediating chondrocyte proliferation by inhibiting chondrocyte hypertrophy which maintains chondrocyte proliferation (missing references).
Row 184. “researches” is incorrect please consider using “studies”.
Row 195. Authors cite Kobayashi et al in 2019… it is in fact Grigelioniene et al in 2019. I noticed there are several references whether the authors cite using the family name of the study lead instead of the first author. Is this the recommendation of the Editorial Office? I have not seen this type of citations previously but maybe this is something new.
Table 1. The authors do not specify what is important/has been shown in human studies, and what has been shown in experimental animal models. Since rodents do not close growth plate, there might be different mechanisms regulating longitudinal bone growth and closure of epiphyseal growth plate. For example most of microRNA studies have been done in experimental animal models, and this should be annotated in the Table 1 “Important microRNAs involving…”
In Table 1 miR-140 is missing.
Row 218. In 2018, Baron et al. Showed, should be corrected to …Jee et al
Row 284. Samd family and cartilage development should be spelled Smad family.
Rows 414-416 “Numerous diseases affecting cartilage are called chondropathies and can be divided into the following subgroups involving Achondroplasia, cartilage tumors, costochondritis, OA, spinal disc, herniation, and relapsing polychondritis”. From the clinical point of view there are more than 400 known entities of skeletal dysplasia and it is totally unclear why authors are mentioning achondroplasia, which is a single entity. After that authors mention cartilage hair hypoplasia, which is of course important and is caused by mutations in RNA coding gene. The authors should have emphasized that.
There are other syndromes caused by haploinsufficiency of microRNA complexes such as Feingold syndrome type 2, and the mechanisms behind it are not even discussed in this article.
Row 443. Using novel techniques such as microarray analysis, MD et al. identified lncRNAs that were. Is MD an author name. Please double check.
Row 457. “in molecular treating”. What is molecular treating?? All medicines are molecules and what do the authors mean by specifically using this term?
Figure 2, legend: heads of the bones should be changed to epiphyses.
Author Response
Dear Editor,
We appreciate the meticulous reviews and constructive suggestions which make the manuscript more acceptable. It is our belief that the manuscript is substantially improved after making the suggested edits.
We amended the manuscript point by point according to reviewers’ comments and we resubmitted the manuscript with track changes. The revision has been developed in consultation with all co-authors, and each author has given approval to the final form of this revision.
Thanks for very much your kind consideration.
Sincerely,
Dr. Sadegh Babashah, Ph.D., Department of Molecular Genetics, Faculty of Biological Sciences, Tarbiat Modares University, P.O. Box: 14115-154, Tehran, Iran, Tel.: +98 21 82884468, Fax: +98 21 82884717, Email: sadegh.babashah@gmail.com; babashah@modares.ac.ir
Dr. William Chi-shing Cho, Department of Clinical Oncology, Queen Elizabeth Hospital, Kowloon, Hong Kong; Tel: +852-35066284, Fax: +852-35065455; Email: williamcscho@gmail.com; chocs@ha.org.hk
— Reviewer 1# — |
“The flow and the logic of it is not always easy to understand and the manuscript looks like a collection of spread facts/data summaries which seem to be difficult to be put together as pieces of one puzzle. The logics and flow of the presented facts need some revision. Abbreviations are used without first time spelling out. English needs some minor revision.
We corrected the language errors. As well as, all abbreviations were double-checked and the only first-used abbreviations were removed. We revised flow of the presented facts and given logics; for this purpose, we removed the unrelated subjects as well as adding new ones! Furthermore, we inserted a novel title to the existed text “The Role of miRNAs in Cartilage Homeostasis” and thus isolated the text from the unrelated context. We tried to unify the text by applying cohesion and coherence.
……………………………………………………………………………..………………………………
“Authors emphasize that microRNAs could be used as therapy for osteoarthritis, or cartilage disease, however, do not inform about pitfalls of this eventual therapy. I think that overall they are too optimistic considering that microRNAs regulate many more genes, than it was reported by earlier studies. Current techniques such as transcriptome studies have demonstrated that many more genes are involved than previously thought, indicating that this therapy may lead to unexpected side effects.”We added two paragraphs to “Therapeutics of ncRNAs of the cartilage diseases” section to address these issues.
…………………………………………………………………………………………………..…………
“Overall through the review article the authors do not separate on data obtained using human cells, experimental animals, or human subjects. This is an important issue because there are important differences in experimental animals’ vs humans in the different processes involved in endochondral bone formation.”We added the source of cartilage or other cells applied by demonstrating the cell source, e.g mouse, Human, rat, etc. additionally, in table 1, we added a column to show the reliable resources that cells had been derived.
……………………………………………………………………………………………………………
“Row 72. Chondrodysplasia generally is not so commonly used term anymore; I recommend changing it to skeletal dysplasia.”Thank you for your suggestion. We replaced “Chondrodysplasia” with “skeletal dysplasia.”
……………………………………………………………………………………………………………
Rows 117-118. “Different genes and transcription factors promote the maturation of MSCs to mature chondrocyte cells (missing references). “Different genes code for transcription factors and other important proteins in chondrogenesis thus I think it is important to use “different molecules promote” …We changed “Different genes and transcription factors” to “Different molecules and encoded proteins, especially transcription factors, promote…”.
……………………………………………………………………………..………………………………
“Recent studies have demonstrated that transcriptional co-regulators significantly contribute to the epigenetic regulatory system (missing references).”We added a new reference for this claim: “Quan, Z., Zheng, D. and Qing, H., 2017. Regulatory roles of long non-coding RNAs in the central nervous system and associated neurodegenerative diseases. Frontiers in cellular neuroscience, 11, p.175.”
………………………………………………………………………………………………………..……
“Rows 147 – 148. Various other reports indicate that miR-140 plays a role in chondrogenesis by indirectly mediating chondrocyte proliferation by inhibiting chondrocyte hypertrophy which maintains chondrocyte proliferation (missing references).”
We added two related references for this sentence.
…………………………………………………………………………………………………..…………
Row 184. “researches” is incorrect please consider using “studies”.Thank you for your comment. We emended them.
……………………………………………………………………………………………………………..
“Row 195. Authors cite Kobayashi et al in 2019… it is in fact Grigelioniene et al in 2019. I noticed there are several references whether the authors cite using the family name of the study lead instead of the first author. Is this the recommendation of the Editorial Office? I have not seen this type of citations previously but maybe this is something new.”Yes, it was a fault; we mistakenly considered the last name of corresponding author(s)’ name. I amended all of authors mentioned in the manuscript according to this comment.
……………………………………………………………………………………………………………..
“Table 1. The authors do not specify what is important/has been shown in human studies, and what has been shown in experimental animal models. Since rodents do not close growth plate, there might be different mechanisms regulating longitudinal bone growth and closure of epiphyseal growth plate. For example, most of microRNA studies have been done in experimental animal models, and this should be annotated in the Table 1 “Important microRNAs involving…”
We added the cell-source of applied studies, e.g mice, Human, rat, etc. Additionally, in table 1, we added a column to show the reliable resources that cells had been derived.
……………………………………………………………………………………………..………………
“In Table 1 miR-140 is missing.”As we mentioned, in table 1, we tried to summarize some of the new miRs with established roles in chondrogenesis or chondrocyte development. However, according to this comment, we added miR-140 to table 1.
………………………………………………………………………………………………..……………
Row 218. In 2018, Baron et al. Showed, should be corrected to …Jee et alBased on the previous comments, we corrected all of the names mentioned in this manuscript.
…………………………………………………………………………………………………..…………
Row 284. Samd family and cartilage development should be spelled Smad family.We corrected this one.
……………………………………………………………………………………………………..………
Rows 414-416 “Numerous diseases affecting cartilage are called chondropathies and can be divided into the following subgroups involving Achondroplasia, cartilage tumors, costochondritis, OA, spinal disc, herniation, and relapsing polychondritis”. From the clinical point of view there are more than 400 known entities of skeletal dysplasia and it is totally unclear why authors are mentioning achondroplasia, which is a single entity. After that authors mention cartilage hair hypoplasia, which is of course important and is caused by mutations in RNA coding gene. The authors should have emphasized that. There are other syndromes caused by haploinsufficiency of microRNA complexes such as Feingold syndrome type 2, and the mechanisms behind it are not even discussed in this article.Thank you for your clarifying. Actually, in table 3 we just focused on lncRNAs and their role in human diseases, not miRs!
Secondly, we tried to concentrate on “Disorders of Early Differentiation” in which chondrocytes cogently play roles. For this purpose, we added a row to table 3 to insert Brachydactyly type E which was determined that has a great association with a lncRNA. The second group we focused was “Defects of Joint Formation” in which osteoarthritis is eminent. To caveat, we researched around 135 syndromes, disorders, and diseases with a chondropathy background, however, the mentioned diseases in table 3 had an association/epigenetic basis related to lncRNAs.
Last but not least, as you mentioned, Feingold syndrome type 2 has an epigenetic cause in which six miRs take a center stage, but there is not any evidence of lncRNA’s embroilments. Thus, we did not insert this syndrome to table 3.
……………………………………………………………………………………………………..………
Row 443. Using novel techniques such as microarray analysis, MD et al. identified lncRNAs that were. Is MD an author name. Please double check.We thoroughly amended the names of the mentioned authors in this study. We had included the studies by their corresponding authors, not their first authors, but in the revised version, we corrected all of them.
……………………………………………………………………………………………………..………
Row 457. “in molecular treating”. What is molecular treating?? All medicines are molecules and what do the authors mean by specifically using this term?This sentence was so perplexing thus we amended it by replacing “In molecular treating” with “OA treatment”.
……………………………………………………………………………………………………..………
Figure 2, legend: heads of the bones should be changed to epiphyses.We changed it according to this comment.
……………………………………………………………………………………………………..………

Reviewer 2 Report
The manuscript entitled “Non-Coding RNAs in Cartilage Development: An Updated Review” is relatively well designed, well written and properly organized. This manuscript has a potential to become an interesting and important paper for the researchers working in bone biology. I have some comments about the manuscript, which I hope, may help them to improve their interpretations.
Major comments:
1. The font size in the Figures 1 and 2 is too small to read; or the resolution of these images is very low. In addition, the authors did not provide a sufficient explanation for these figures.
2. In lines 124-125, the authors declare that “Regarding epigenetic factors, it is obvious that miRNA and lncRNA play an important role in chondrogenesis”. Although this sentence is used as a summary at the end of the paragraph of “3. Non-coding RNAs and chondrocyte differentiation”, there is no evidence about it in the paragraph. The authors should cite well-chosen references and have to explain logically. In addition to this sentence, there are other sentences or paragraphs that need to be corrected for the same reason in the text.
3. What kind of order is the molecules in the Tables 1-3? Most molecules are not mentioned in the text. Additionally, the authors should unify the style of words in “Target” and “Effects” of these tables, because some are nouns and some are verbs……. Furthermore, the words in “study models” of Table 2 are vague.
Minor comments:
1. Even if some technical terms appear only once in the text, the authors wrote abbreviated words such as RISC (RNA-induced silencing complex), HMT (histone methyltransferase), GO (gene ontology), EWAS (epigenome-wide association study), and etc. Because these abbreviated words are less common, it makes hard to read; then these abbreviated words should be deleted.
2. What is “PTM” in line 130?
Author Response
Dear Editor,
We appreciate the meticulous reviews and constructive suggestions which make the manuscript more acceptable. It is our belief that the manuscript is substantially improved after making the suggested edits.
We amended the manuscript point by point according to reviewers’ comments and we resubmitted the manuscript with track changes. The revision has been developed in consultation with all co-authors, and each author has given approval to the final form of this revision.
Thanks for very much your kind consideration.
Sincerely,
Dr. Sadegh Babashah, Ph.D., Department of Molecular Genetics, Faculty of Biological Sciences, Tarbiat Modares University, P.O. Box: 14115-154, Tehran, Iran, Tel.: +98 21 82884468, Fax: +98 21 82884717, Email: sadegh.babashah@gmail.com; babashah@modares.ac.ir
Dr. William Chi-shing Cho, Department of Clinical Oncology, Queen Elizabeth Hospital, Kowloon, Hong Kong; Tel: +852-35066284, Fax: +852-35065455; Email: williamcscho@gmail.com; chocs@ha.org.hk
Reviewer 2# |
Thank you for your comment. We amended the figures according to this rational comment. It also should be said that the figures attached in the main manuscript have not sufficient resolution. Although we amended the figures’ font size, we requested from the editor to import the main original submitted figures which are in a high resolution. Besides, we added some parts as figure legends in this manuscript which will help readers to understand the context.
……………………………………………………………………………………………………..........…
“In lines 124-125, the authors declare that “Regarding epigenetic factors, it is obvious that miRNA and lncRNA play an important role in chondrogenesis”. Although this sentence is used as a summary at the end of the paragraph of “3. Non-coding RNAs and chondrocyte differentiation”, there is no evidence about it in the paragraph. The authors should cite well-chosen references and have to explain logically. In addition to this sentence, there are other sentences or paragraphs that need to be corrected for the same reason in the text”This paragraph was a general paragraph introducing a new subject namely ncRNAs and their roles in chondrogenesis which were pursued in following in form of these subtitles: “miRNA and chondrocyte differentiation”, “Other important miRNAs contributing to chondrocyte differentiation”, “Role of lncRNAs in chondrogenesis”. However, because this paragraph was so intriguing for readers, we changed it thoroughly and removed the title. In fact, we integrated the text to the previous paragraph.
…………………………………………………………………………………………...……………...…
“What kind of order is the molecules in the Tables 1-3? Most molecules are not mentioned in the text. Additionally, the authors should unify the style of words in “Target” and “Effects” of these tables, because some are nouns and some are verbs……. Furthermore, the words in “study models” of Table 2 are vague.”Actually, in this table, we tried to summarize the newest miRs which have a great association with chondrogenesis or chondrocyte development. Except for miR-140, all miRs are detected or introduced in either 2018 or 2019. Many of them are novel and there is a lack of comprehensive information about the exact role of them in chondrogenesis. So, we just enumerated them and not imported in the text. On the other hand, in the text we discussed some of the well-established and well-known studies. Secondly, we amended the targets and Effects section in this table by considering a unique writing style.
Thirdly, in table 2, we tried to distinguish the results of applied studies by considering the type of their experimental. For example, MSC is a well-established cell model. To mitigate the unwanted misunderstanding, we change the “Study Models” to “cell type.”
………………...………………………………………………………………………………………...…
“Even if some technical terms appear only once in the text, the authors wrote abbreviated words such as RISC (RNA-induced silencing complex), HMT (histone methyltransferase), GO (gene ontology), EWAS (epigenome-wide association study), and etc. Because these abbreviated words are less common, it makes hard to read; then these abbreviated words should be deleted.”We double-checked the text and expunged the only first used abbreviation as well as the mentioned ones in this comment.
………………………………………………………………………………………………………...……
What is “PTM” in line 130?PTM is a short form of post-transcriptional modification and according to the previous comment, we emended it by removing its abbreviation form which was used for the first time.

Round 2
Reviewer 1 Report
Row 210. "dwarf phenotype" should be removed (miR40140 KO show only mild short stature). “Dwarf” is a term which by skeletal dysplasia experts is no longer recommended to use to describe disease status in humans. I think this is also not correct term for animals, because it can be easily exchanged to “mild, moderate or severe short stature” depending on the discussed condition.
Row 223. what do the authors mean by “via their impressions on BMP7 and TGF-β2”?
In table 1. “resource” should be changed to cell type in accordance with Table 2. In the section on miR107 “modules” should be spelled instead “modulates”
Achondroplasia and Human cartilage hair hypoplasia can be spelled using lower cases in the middle of the sentence.
Row 479. “These mutations impress… should be spelled “These mutation affect…”
Row 482 should be spelled “…patients affected by cartilage hair hypoplasia…”
Row 486. “This study can show the lncRNAs efficiency to treat cartilage diseases”. The statement is wrong, the study only shows that molecular mechanisms/consequences of CHH, but not that lncRNA can be used for treatment of CHH.
Row 489. The statement “gain-of-function” is wrong. The mutation described in that study leads to loss of normal function and gain-of-new-function in miR140, eg mutation is neomorphic (which is functionally different than simple gain-of-function).
Row 490. The statement “They demonstrated a delayed cartilage maturation of the larynx, trachea, and anterior ribs in the mouse model” is wrong. The study describes delayed maturation of growth plate chondrocytes all over the skeleton, as well as delayed maturation of the epiphyses in mice. It also showed that the above described abnormalities are present in humans affected by novel spondyloepiphyseal dysplasia and recapitulated in the transgenic mice model.
Rows 540-542 contain a strange expression: what is a “personalized therapeutic biomarker”? In clinical point of view, biomarkers are not medicine or therapeutic agents, biomarkers are factors in tissues or blood indicated the disease status. The sentence in these rows has to be corrected/removed.
Author Response
Dear editor,
Thank you for the opportunity to revise our manuscript. We appreciate the careful review and constructive suggestions which make the manuscript more acceptable. It is our belief that the manuscript is substantially improved after making the suggested edits.
We amended the manuscript point by point according to the reviewer's comments. Following this letter are the editor and reviewers’ comments with our responses in italics, including how and where the text was modified. According to your suggestion, changes are visible in “Track Changes”. The revision has been developed in consultation with all co-authors, and each author has given approval to the final form of this revision.
Row 210. "dwarf phenotype" should be removed (miR40140 KO show only mild short stature). “Dwarf” is a term which by skeletal dysplasia experts is no longer recommended to use to describe disease status in humans. I think this is also not the correct term for animals, because it can be easily exchanged to “mild, moderate or severe short stature” depending on the discussed condition.Thank you for this recommendation. We substituted the “dwarf phenotype” with “Short stature”.
………………………………………………………………………………………………………………
Row 223. what do the authors mean by “via their impressions on BMP7 and TGF-β2”?We changed this sentence to “through BMP7 and TGF-β2R.”
………………………………………………………………………………………………………………
In table 1. “resource” should be changed to cell type in accordance with Table 2. In the section on miR107 “modules” should be spelled instead “modulates”Thank you for your meticulous comment. We corrected it.
………………………………………………………………………………………………………………
Achondroplasia and Human cartilage hair hypoplasia can be spelled using lower cases in the middle of the sentence.We change the format of these names to lowercase.
………………………………………………………………………………………………………………
Row 479. “These mutations impress… should be spelled “These mutations affect…”We replaced “impress” with “affect”.
………………………………………………………………………………………………………………
Row 482 should be spelled “…patients affected by cartilage hair hypoplasia…”We corrected it.
………………………………………………………………………………………………………………
Row 486. “This study can show the lncRNAs efficiency to treat cartilage diseases”. The statement is wrong, the study only shows that molecular mechanisms/consequences of CHH, but not that lncRNA can be used for treatment of CHH.We changed this sentence to “This study suggested that alteration of ribosomal processing in CHH is …” which shows the direct consequences derived from this study.
……………………………………………………………………………………………………………
Row 489. The statement “gain-of-function” is wrong. The mutation described in that study leads to loss of normal function and gain-of-new-function in miR140, eg mutation is neomorphic (which is functionally different than simple gain-of-function).To clarify, we changed the sentence to “gain of a new function of miR-140.”
……………………………………………………………………………………………………………
Row 490. The statement “They demonstrated a delayed cartilage maturation of the larynx, trachea, and anterior ribs in the mouse model” is wrong. The study describes delayed maturation of growth plate chondrocytes all over the skeleton, as well as delayed maturation of the epiphyses in mice. It also showed that the above described abnormalities are present in humans affected by novel spondyloepiphyseal dysplasia and recapitulated in the transgenic mice model.We changed the sentence according to this comment.
……………………………………………………………………………………………………………
Rows 540-542 contain a strange expression: what is a “personalized therapeutic biomarker”? In clinical point of view, biomarkers are not medicine or therapeutic agents, biomarkers are factors in tissues or blood indicated the disease status. The sentence in these rows has to be corrected/removed.Thank you for this recommendation. We altered the misleading sentence.
……………………………………………………………………………………………………………

Reviewer 2 Report
The author provided answers that I have requested; therefore this manuscript is enough to accept IJMS.
Author Response
Dear Editor
Thank you very much for your consideration and attention.
Sincerely,
Sadegh Babashah, Ph.D., Department of Molecular Genetics, Faculty of Biological Sciences, Tarbiat Modares University, P.O. Box: 14115-154, Tehran, Iran, Tel.: +98 21 82884468, Fax: +98 21 82884717, Email: sadegh.babashah@gmail.com; babashah@modares.ac.ir
William Chi-shing Cho, Department of Clinical Oncology, Queen Elizabeth Hospital, Kowloon, Hong Kong; Tel: +852-35066284, Fax: +852-35065455; Email: williamcscho@gmail.com; chocs@ha.org.hk